# Therapeutic Advances in Initially Unresectable Locally Advanced Intrahepatic Cholangiocarcinoma: Emerging Treatments and the Role of Liver Transplantation

**DOI:** 10.3390/curroncol32060293

**Published:** 2025-05-22

**Authors:** Sofia Lopiano, James V. Guarrera, Keri E. Lunsford

**Affiliations:** Division of Transplant and HPB Surgery, Department of Surgery, Rutgers New Jersey Medical School, Newark, NJ 07103, USA; sl2173@njms.rutgers.edu (S.L.); james.guarrera@njms.rutgers.edu (J.V.G.)

**Keywords:** intrahepatic cholangiocarcinoma, neoadjuvant chemotherapy, transarterial chemoembolization, transarterial radioembolization, hepatic artery infusion therapy, liver transplantation

## Abstract

Optimal curative therapy for intrahepatic cholangiocarcinoma (iCCA) involves hepatic resection; however, due to its insidious nature, iCCA frequently presents at advanced stages. Consequently, 70–80% of patients feature unresectable iCCA at presentation. Recent expansions in therapeutic options for locally advanced unresectable iCCA include immunotherapy, targeted chemotherapeutics, and liver-directed therapies. These have increased progression-free survival, enhanced response rates, and improved downstaging for resection. Liver transplant has also emerged as an alternative for patients whose tumors remain unresectable despite therapeutic response. Here, we explore emerging treatment options included in a multidisciplinary treatment paradigm to prolong survival in patients with initially unresectable locally advanced iCCA.

## 1. Introduction

Cholangiocarcinoma is a highly aggressive malignancy arising from the biliary epithelium, and it is classified according to its anatomic site of origin [1]. Intrahepatic cholangiocarcinoma (iCCA) arises from segmental and distal ducts within the liver parenchyma. Extrahepatic cholangiocarcinoma subtypes include perihilar cholangiocarcinoma (pCCA), originating from the right, left, and/or common hepatic duct, and distal cholangiocarcinoma (dCCA), originating from the common bile duct distal to the insertion of the cystic duct [2,3]. Intrahepatic CCA is further classified by a growth pattern into mass-forming, periductal infiltrating, and intraductal subtypes [3]. Mass-forming iCCA is the most common, with a prevalence of 60%, while 20% of cases are classified as either periductal or mixed type [4].

CCA accounts for 10–15% of liver cancers, and iCCA is the least common subtype (10–20% are CCA) [5]. The incidence of iCCA in the United States, however, has steadily increased from 0.49 to 1.48 per 100,000 persons from 2000 to 2019 [6]. Median overall survival (OS) has recently improved but remains dismal (from 5 months in 2000 to 9 months in 2017) [6]. The best outcomes for iCCA are achieved with resection in the absence of distant lymph nodes or metastatic disease, and long-term survival for patients with initially unresectable locally advanced disease has been historically limited. In the present review, we explore emerging treatment options included in a multidisciplinary treatment paradigm to prolong survival in patients with initially unresectable locally advanced iCCA.

## 2. Pathogenesis, Presentation, and Diagnosis

Risk factors for iCCA often relate to biliary inflammation, which includes parasitic infection, choledochal cyst, chronic biliary disease, recurrent pyogenic cholangitis, hepatitis B or C, liver flukes, and toxin exposure (asbestos, dioxins, or nitrosamines) [2,6]. Metabolic diseases such as diabetes, obesity, metabolic dysfunction associated with steatotic liver disease (MASLD), and cirrhosis also increase the risk, although iCCA can arise in either cirrhotic or non-cirrhotic livers [2].

Diagnosis of iCCA often occurs at late stages due to its lack of early symptoms and limited detection through non-invasive methods [5,7]. While pCCA and dCCA often present with jaundice due to biliary obstruction, iCCA rarely presents with jaundice [7]. At advanced stages, iCCA presents with non-specific symptoms, including abdominal pain, malaise, nausea, anorexia, and weight loss [7]. Early-stage diagnosis is usually incidental (20–25%) during imaging performed for screening or other indications [7]. Blood biomarkers, including CA19-9 and CEA, can be evaluated, but their specificity is limited as benign liver pathologies may also increase biomarker levels [4].

High-quality cross-sectional imaging, such as contrast-enhanced multiphasic CT, MRI, and MRCP, is necessary for the staging and assessment of primary tumor location, vascular invasion, lymph node involvement, and distant metastases [7]. On CT, iCCA appears as a well-defined or infiltrative hypodense hepatic lesion, which may demonstrate distal biliary dilatation or capsular retraction [4]. Due to its portal blood supply, portal-phase or delayed-phase enhancement may distinguish it from hepatocellular carcinoma (HCC). The mass may also demonstrate peripheral arterial hyperenhancement, with centripetal enhancement in later phases [8]. On MRI or MRCP, iCCAs appear as hypointense lesions on T1- and heterogeneously hyperintense lesions on T2-weighted images [4]. The lesion may demonstrate peripherally restricted diffusion, with less restriction in central areas. Gadoxetic acid-enhanced MRI may distinguish between mixed HCC-CCA and mass-forming iCCA, as iCCA may demonstrate a target-shaped enhancement [4,8].

Unfortunately, the sensitivity of MRI or CT alone in the detection of distant lymph node metastases is quite low. While FDG-PET is not routinely used to stage iCCA, it may be useful to identify occult metastases. This may inform candidacy for resection or transplant [4,8]. EUS/ERCP may supplement evaluation for intervention on biliary obstruction and EUS-guided aspiration of enlarged lymph nodes. Ultimately, diagnostic laparoscopy is recommended for evaluation of peritoneal and lymph node involvement prior to surgical intervention.

Histopathologic or cytologic analysis is generally required to confirm the diagnosis. Brush cytology or biopsy can be performed using ERCP or PTC (diagnostic sensitivities of 56% for brushings, 67% for biopsy, and 70.7% for biopsy+brushings). Alternatively, EUS-guided fine needle aspiration (FNA) can be employed; however, it is more effective for extrahepatic CCA, and concern for peritoneal seeding limits its utility [4]. For iCCA, image-guided core needle biopsy is preferred, as it provides a larger diagnostic sample sufficient for additional genetic analysis [9]. For all biopsy samples, next-generation sequencing should be performed to assess eligibility for immunotherapy and targeted systemic therapy options.

## 3. Surgery as the Gold Standard for iCCA Treatment

Resection with negative microscopic margins is the only proven curative option for iCCA. Unfortunately, only a third of patients presenting with iCCA are eligible for resection [10]. Five-year overall survival (OS) following resection is poor, at 20–35%, with recurrence-free survival (RFS) at 2–39%, with a median OS of 28 months [1,11]. While tumor number, vascular invasion, and lymph node involvement are prognostic indicators for patients following resection, tumor size is not a significant prognostic factor [12]. Prognosis following R0 resection differs based on TNM stages, with a 5-year OS of 45.3%, 28.9%, 13.7%, and 0% for TNM stages I, II, III, and IV, respectively [13]. Thus, for patients with locally advanced or initially unresectable iCCA, alternative or combined treatment modalities should be considered.

Patients whose tumors can be resected with negative histologic margins while maintaining an adequate liver remnant should be offered upfront resection. An adequate future liver remnant (FLR) requires a minimum of two contiguous segments with sufficient perfusion and both venous and biliary drainage. Underlying parenchymal dysfunction further defines the required volume and is estimated at 20% of the total liver volume for a normal liver, 30% for an injured one, and ≥40% for fibrotic or cirrhotic liver. Approaches to increase the FLR include portal vein embolization, combined portal and hepatic vein embolization, and Y90. Extrahepatic disease (including lymph nodes beyond the regional basin) is a contraindication to resection. Bilateral multifocal or multicentric disease also has poor outcomes following resection [9]. Evidence supporting the survival benefit of R1 resections is lacking [14,15]. Data is emerging to suggest that initially unresectable iCCA, especially in the setting of liver-limited disease, may be downstaged to allow for resection.

Adjuvant therapy following curative intent resection in iCCA has proven efficacy. The BILCAP trial evaluated oral capecitabine following biliary tract cancer (BTC) resection, demonstrating a median OS of 51.1 months with adjuvant capecitabine compared to 36.4 months in observation. RFS was also improved to 24.4 months with capecitabine versus 17.5 months in observation [16]. Trials evaluating alternative adjuvant regimens include the PRODIGE 12 trial. Patients were randomized to either GEMOX (gemcitabine + cisplatin) or surveillance following R0 or R1 resection for BTC. RFS with GEMOX was improved to 30.4 months compared to 18.5 months with observation, but the results were not statistically significant [17]. The ESPAC-3 reported a statistically significant OS benefit with adjuvant fluorouracil or gemcitabine following resection for pancreatic and BTC [18]. Based on the findings from the BILCAP trial, the ASCO clinical practice guideline currently recommends adjuvant capecitabine for 6 months following BTC resection [19].

## 4. Systemic Chemotherapy for Initially Unresectable iCCA

First-line treatment for locally advanced and unresectable iCCA is systemic chemotherapy, which can downstage for resection, treat micrometastatic disease, and halt disease progression. The ABC-02 trial established a significant survival benefit to using gemcitabine and cisplatin (Gem/Cis) combination therapy for advanced BTC, leading to its adoption as the standard first-line regimen [20]. Gem/Cis remained the standard treatment until the publication of the TOPAZ-1 trial in 2022. TOPAZ-1 evaluated durvalumab (a monoclonal antibody against PD-L1) plus Gem/Cis (Gem/Cis/Durva) versus Gem/Cis+placebo, in which 685 patients with unresectable, metastatic, or recurrent BTC were randomized. Twenty-four-month OS with Gem/Cis/Durva was 24.9% compared with 10.4% for Gem/Cis alone [21]. Pembrolizumab (a monoclonal antibody against PD-1) in combination with Gem/Cis (Gem/Cis/Pembro) was assessed in the KEYNOTE-966 trial. Median OS with Gem/Cis/Pembro was 12.7 months compared with 10.9 months for Gem/Cis alone (*p* = 0.0034) with similar toxicity [22]. This evidence supports adding immunotherapy to Gem/Cis in the first-line setting. An alternative combination of Gem/Cis and nab-paclitaxel (GAP) has been concurrently evaluated for systemic treatment for advanced BTC. Significant enthusiasm for the combination arose from the single-arm, phase 2 results, which demonstrated improved progression-free survival (PFS) (11.8 months vs. 8 months) and prolonged OS (19.2 months vs. 11.2 months) compared to historical Gem/Cis controls [23]. This prompted the phase III SWOG1815 trial, which randomized 441 advanced BTC patients to either GAP or standard Gem/Cis dosing. While there was no overall statistically significant improvement in OS (14 months vs. 12.7 months, respectively), select patient groups, including those with locally advanced disease, did exhibit some benefit [24].

Neoadjuvant for downstaging: With improved disease response observed in TOPAZ and SWOG for patients with locally advanced iCCA, there has been a resurgent interest in neoadjuvant chemotherapy for downstaging prior to liver resection. Neoadjuvant chemotherapy has the potential to downstage tumors, control micrometastatic disease, eradicate lymph node metastases, and improve the probability of achieving R0 resection [25]. Current evidence regarding the use of neoadjuvant treatment largely comes from retrospective and single-case analyses [26]. A propensity-matched retrospective analysis of 1450 patients demonstrated superior OS with neoadjuvant versus adjuvant chemotherapy [25]. Another single-institution retrospective analysis demonstrated prolonged OS with resection with or without neoadjuvant therapy for iCCA compared to chemotherapy alone (24.1, 25.7, and 7.8 months, respectively). While neoadjuvant therapy did not significantly prolong OS compared to surgery alone, 39 of 74 patients receiving neoadjuvant therapy had initially unresectable locally advanced iCCA. Thus, initially unresectable patients who are successfully downstaged demonstrate similar OS and RFS to patients whose tumor is initially resectable [27].

Prospective studies evaluating neoadjuvant therapies for cholangiocarcinoma are underway. A recent multi-institutional, phase II trial evaluating neoadjuvant gemcitabine, cisplatin, and nab-paclitaxel (NEO-GAP) for resectable high-risk intrahepatic cholangiocarcinoma demonstrated that this strategy is feasible and not associated with significant postoperative morbidity. All 30 enrolled patients completed preoperative chemotherapy, and 22 patients completed resection per protocol. Of 30 enrolled patients, 90% exhibited disease control, with 23% demonstrating a partial response. A total of 73% of patients completed therapy and progressed to resection. Median OS for the entire cohort was 24 months, with recurrence in 43% of patients following surgery [28]. Thus, neoadjuvant regimens may improve outcomes for patients with initially unresectable disease.

Given the demonstrated feasibility, safety, and efficacy of neoadjuvant treatment in the NEO-GAP trial, additional neoadjuvant trials evaluating combined chemotherapy and immunotherapy or targeted therapy have been initiated. Building off the NEO-GAP study, the upcoming OPT-IC trial (NCT03579771) will perform a phase II feasibility and safety assessment of neoadjuvant gemcitabine, cisplatin, and nab-paclitaxel, with an FGFR2 inhibitor added for patients with FGFR2 fusion or rearrangements [29]. The NCI’s Experimental Therapeutics Clinical Trials Network (ETCTN) is conducting a phase II trial evaluating neoadjuvant durvalumab with gemcitabine/cisplatin chemotherapy in patients with high-risk, resectable iCCA (ETCTN-10608, NCT06050252). Primary objectives are evaluation of treatment drop-out and completion of therapy/resection, with a secondary objective of determining pathologic response [30]. MD Anderson Cancer Center is similarly conducting a single-arm, phase II trial assessing pembrolizumab in combination with gemcitabine/cisplatin for borderline resectable iCCA, with RFS and major pathologic response as primary endpoints (NCT05967182) [31]. Finally, the NEOLANGIO trial (NCT06569225) at the University of Toronto will evaluate neoadjuvant gemcitabine, cisplatin, and nab-paclitaxel, along with rilvegostomig (an anti-TGIT/anti-PD-1 bispecific antibody) in resectable iCCA [32]. Collectively, these efforts signal a growing interest in optimizing neoadjuvant strategies through the incorporation of immunotherapy, with the goal of enhancing surgical outcomes and overall survival. Prospective trials evaluating neoadjuvant regimens for iCCA are summarized in Table 1.

Targeted therapies: Cholangiocarcinoma exhibits substantial genetic heterogeneity, making next-generation sequencing valuable for identifying potential drug targets and predicting prognosis [33]. The most common mutations include IDH1, ARID1A, BAP1, TP53, and FGFR2 gene fusions [33]. FGFR2 gene arrangements and gain-of-function mutations in IDH1 are actionable through targeted treatment. Such targets are the subject of extensive clinical investigation as first- and second-line therapy [33]. Among cholangiocarcinoma patients with FGFR fusions or rearrangements, 35.5% achieved an objective response when treated with second-line pemigatinib, an oral inhibitor of FGFR1, FGFR2, and FGFR3 [34]. FGFR2 fusion- or rearrangement-positive cholangiocarcinoma patients treated with futibatinib, a covalent FGFR inhibitor, also experience clinical benefit [35]. Ivosidenib, an IDH-1 inhibitor, demonstrates prolonged OS and PFS in patients with IDH-1 mutant unresectable disease [20,36]. These findings highlight the clinical importance of genetic profiling to identify actionable mutations, and indicate that targeted therapies may offer future benefit for downstaging patients with locally advanced unresectable disease.

## 5. Use of Locoregional and Liver-Directed Therapies in Initially Unresectable iCCA

Locoregional therapy is a cornerstone of treatment in the setting of unresectable HCC, and such liver-directed options may add benefit to the treatment of locally advanced iCCA.

Radiation therapy: Advancements in radiation delivery and safety have driven interest in its use for advanced iCCA [37]. In a single-institution retrospective study, patients with inoperable iCCA treated with high-dose radiation demonstrated survival rates comparable to those previously reported with resection [38]. Proton beam therapy, a form of external beam radiation therapy (EBRT), has also shown promise. In a phase II clinical trial, treatment with high-dose hypofractionated proton therapy improved tumor control and survival in patients with unresectable iCCA [37]. Proton beam therapy is hypothesized to lessen unwanted radiation exposure and reduce the risk of radiation-induced liver disease [37]. Pencil beam scanning (PBS) proton beam therapy further decreases doses to nearby organs and reduces radiation-induced injury [39]. Proton beam therapy administered with PBS has been shown to be safe and effective on retrospective analysis for large and multifocal liver tumors, but further investigation is needed to confirm survival benefits [39]. High-dose iCCA radiation alone may be sufficient to improve OS and PFS; however, utility for downstaging, especially at high doses, may complicate and increase the morbidity of operative resection.

TACE: Transarterial chemoembolization (TACE) directly infuses a concentrated chemotherapeutic emulsion and iodized oil via the hepatic arterial system directly into tumor tissue. DEB-TACE delivers chemotherapy using drug-eluting beads rather than iodized oil. Localized delivery of chemotherapeutic agents minimizes hepatic and systemic toxicity [40]. While TACE has been extensively studied in HCC, evidence supporting its use for iCCA is primarily based on retrospective studies, particularly in unresectable disease [41]. For unresectable iCCA, retrospective analysis reveals prolonged overall survival with TACE [42,43,44,45]. Multicenter prospective phase II evaluation demonstrates improved OS in unresectable iCCA following combined drug-eluting irinotecan (DEBIRI) TACE combined with systemic Gem/Cis compared to Gem/Cis alone (33.7 vs. 12.6 months). Combined DEBIRI treatment additionally resulted in greater downstaging (25%) compared to Gem/Cis (8%) [45]. Evaluating prognostic factors, including tumor vascularity, initial response to TACE, and Child–Pugh classification, is important before initiating TACE to optimize patient outcomes [44].

TARE: Transarterial radioembolization (TARE) is gaining recognition as a valuable palliative and downstaging option for unresectable iCCA. Transarterial Y90 radioembolization uses beta-emitting microspheres to deliver selective intratumor radiation via hepatic arterial access [46]. A systematic review encompassing 21 predominantly retrospective studies evaluated demonstrated an PFS of 7.8 months and a pooled OS of 12.7 months with TARE [47]. The MISPHEC trial was the first published prospective trial investigating the efficacy of TARE combination with chemotherapy in unresectable iCCA. In 41 patients receiving first-line TARE+Gem/Cis, median OS was 22 months, and PFS was 55% at 12 months and 30% at 24 months [46]. Nine patients (22%) were successfully downstaged to surgical intervention, and eight patients (20%) achieved R0 resection [46]. The sequence of administration of chemotherapy and TARE remains a subject for investigation. In a phase 2 single-arm multicenter study evaluating TARE followed by Gem/Cis, investigators hypothesized that TARE followed by chemotherapy would enhance local control of iCCA prior to systemic control. Among the 16 patients treated, median OS was 21.6 months and median PFS was 9 months. While both clinical trials report a survival benefit to treating with a combination of SIRT and chemotherapy, further investigation is warranted to determine the optimal timing for treatment administration [48].

Hepatic artery infusion pump: Hepatic artery infusion (HAI) therapy delivers high doses of chemotherapy into hepatic arterial circulation using an implanted pump (generally via the gastroduodenal artery). This approach leverages hepatic first-pass metabolism to minimize systemic exposure while maximizing drug delivery to the liver [49,50]. In a comparative meta-analysis, HAI provided superior tumor response and OS compared to TACE, DEB-TACE, and TARE [51]. Retrospective data suggest that a combination of systemic therapy and HAI improves survival compared to systemic chemotherapy alone [52], and current research focuses on the efficacy of combining floxuridine HAI with systemic chemotherapy to maximize tumor response. An MSKCC single-institution phase 2 clinical trial of 38 patients with unresectable iCCA treated with HAI floxuridine plus systemic Gem/Ox demonstrated a PFS of 80% at 6 months with 84% disease control and 58% partial radiographic response at 6 months. Four patients were downstaged to resection. Interestingly, survival benefit was greatest in patients with IDH1/2 mutant tumors (2-year OS 90% vs. 33% for wild-type) [53]. Despite these promising results, large-scale randomized prospective studies to ascertain the true efficacy of HAI therapy are necessary.

Novel chemotherapy combinations with HAI are additionally being investigated. Sun Yat-Sen University Cancer Center (SYSUCC) recently reported that lenvatinib plus durvalumab combined with FOLFOX-HAI achieved a median OS of 17.9 months and a median PFS of 11.9 months, but 46% of patients experienced grade 3 or 4 complications [54]. HELIX-1 (NCT04251714) is an active, single-center, phase II clinical trial investigating the efficacy of the induction of systemic mFOLFIRINOX (folinic acid, fluorouracil, irinotecan, and oxaliplatin) followed by a concurrent HAI of floxuridine and dexamethasone plus systemic mFOLFIRI (folinic acid, fluorouracil, and irinotecan) for unresectable iCCA. Preliminary data presented at ASCO 2024 reported that the regimen is well-tolerated and demonstrates longer disease control compared to historical controls [55]. Future directions include exploring innovative chemotherapy combinations for HAI, conducting phase III trials to evaluate efficacy, and examining the impact of specific disease mutations on treatment outcomes.

Future therapies: While liver-directed therapies have demonstrated disease control and response, combination with emerging systemic treatments may represent the future for iCCA treatment. Recently, a phase II trial reported that external radiation combined with anti-PD1 antibodies for unresectable iCCA is safe and efficacious [56]. Other treatments under current investigation include IL-12 and FUDR HAI in combination with Gem/Ox for unresectable iCCA (NCT05286814) [57]. Finally, histotripsy, a non-invasive non-ionizing and non-thermal ultrasound-based ablation device, has recently been granted FDA clearance. Early studies demonstrate successful ablation of intrahepatic cholangiocarcinoma [58] and, theoretically, the release of tumor antigens through this ablation modality in conjunction with systemic immunotherapy may facilitate immune clearance of tumor cells. These promising results highlight the emerging role of combined liver-directed and systemic therapies in the treatment or downstaging of locally advanced iCCA. OS and PFS for prospective trials evaluating liver-directed therapies are summarized in Table 2.

## 6. Liver Transplantation for Intrahepatic Cholangiocarcinoma

Liver transplant (LT) for unresectable, locally advanced iCCA affords wider surgical margins than resection and treats underlying liver disease [59]. Despite this, LT for iCCA has been historically contraindicated due to poor survival outcomes and high recurrence [60]. A retrospective analysis conducted in 1991 reported 44% recurrence, with a 2- and 5-year OS of 30% and 17% in transplanted iCCA patients [61]. Failure to differentiate between cholangiocarcinoma subtypes, poor patient selection, and lack of neoadjuvant treatment could reduce the applicability of historic outcomes to current practice [62]. Interest in LT for iCCA has expanded with the recent introduction of revised protocols and improved patient selection.

Due to its long-standing contraindication, LT for iCCA studies have largely been retrospective, on patients misdiagnosed with HCC or on tumors discovered incidentally on explant [62]. Several more contemporary retrospective multicenter studies have compared survival and recurrence rates in cirrhotic LT recipients diagnosed with iCCA on explant. In these studies, patients with prior systemic chemotherapy were excluded, and all patients were either untreated or received only locoregional therapy. The first Spanish multicenter study included 29 cirrhotic patients with iCCA, eight of whom had “very early” iCCA (single tumor ≤ 2 cm). While patients with tumors ≥ 2 cm or multinodular tumors had recurrence rates of 36.4%, no patients with “very early” iCCA had a recurrence, and 1-, 3-, and 5-year OS was 100%, 73%, and 73%, respectively [63]. These findings were validated in 2016 through a multinational study by the same group, which reported decreased post-LT recurrence at 1, 3, and 5 years in “very early” iCCA patients (7%, 18%, and 18%) compared with 30%, 47%, and 61% for patients with “advanced disease” [64]. The “very early” iCCA group additionally showed an improved 1-, 3-, and 5-year OS of 93%, 84%, and 65%, compared to 79%, 50%, and 45% for those with “advanced disease” [64]. While these findings were promising, identification of iCCA patients with tumors ≤ 2 cm is rare, given the insidious nature of the disease. A French multicenter retrospective study subsequently compared the outcomes of LT and resection for cirrhotic patients with iCCA or combined HCC/CCA [65]. Overall, LT recipients had an improved 5-year RFS, at 75% compared to 36% for resection. On subgroup analysis, transplanted patients with tumors 2–5 cm had a 21% recurrence rate compared with 48% for those following resection, and a 5-year RFS of 74% compared with 40% for those following resection [65]. These findings suggest LT may be a viable option for cirrhotic patients with tumors ≤ 5 cm.

Due to the rarity of transplant for iCCA, results of retrospective studies have often combined data with the outcomes for hepatocholangiocarcinoma (HCC-CCA). Unlike iCCA, HCC-CCA is thought to arise from hepatic progenitor cells. While both tumor pathologies have been classified as aggressive, HCC-CCA has a distinctly different pathology. These tumors often present in the setting of pre-existing cirrhosis, making tumor resection difficult [66,67,68]. Up to 3% of tumors initially diagnosed as HCC are identified as HCC-CCA on explant. A few studies have separately considered LT outcomes for iCCA. One single-center analysis from UCLA performed propensity matching of patients diagnosed with HCC-CCA (*n* = 12) to patients with HCC (*n* = 36). HCC-CCA tumors were more likely to be poorly differentiated with a higher grade. When matched by explant pathologic criteria (diameter, differentiation, grade, vascular invasion, etc.), HCC-CCA and HCC LT recipients exhibited a comparable 5-year OS and RFS (42% vs. 48% and 42% vs. 44%, respectively), with recurrence limited to patients with poorly differentiated tumors [68]. Data from 12 U.S. transplant centers were subsequently analyzed for patients diagnosed with HCC-CCA compared to those with HCC. Patients with HCC-CCA meeting Milan criteria exhibited higher recurrence (23.1%) compared to those with HCC (11.5%), but 5-year OS was comparable (70.1% HCC-CCA, 73.4% HCC, *p* = 0.806). Given the different tumor biology, results for HCC-CCA should not be combined with those for iCCA as they risk an inadvertent bias of outcomes [69].

Improved survival outcomes for LT in iCCA patients with low tumor burden highlighted the need to clarify the role of neoadjuvant therapy in facilitating LT in locally advanced iCCA. Case series combining neoadjuvant treatment with transplantation (with or without cirrhosis) have also demonstrated a survival advantage in advanced disease. UCLA performed a retrospective analysis comparing LT and resection outcomes for iCCA and pCCA. The 5-year RFS was significantly higher for patients receiving transplant. In the LT group, neoadjuvant plus adjuvant therapy prolonged 5-year RFS to 47%, compared with 20% for no therapy, and 33% for adjuvant therapy alone [70].

Houston Methodist and MD Anderson Cancer Center subsequently led the first prospective case series evaluating neoadjuvant therapy and LT for locally advanced unresectable iCCA. Patients were required to demonstrate >6 months of disease stability on therapy (primarily Gem/Cis) with no restrictions on tumor number or size. The initial six patient cohort demonstrated a 1-, 3-, and 5-year post-transplant OS of 100%, 83.3%, and 83.3%, and an RFS of 50%, respectively [71]. In a 2022 follow-up by the same group, 1-, 3-, and 5-year OS was 100%, 71%, and 57% for 18 patients receiving LT, with an RFS of 72% and 52% at 1 and 3 years [72]. An updated cohort of 26 patients showed similar outcomes, with an OS of 96% at one year and 82.7% at three years, and an RFS of 70.8% and 56.3% at one and three years, respectively [73]. In Norway, the prospective single-arm TESLA trial is also ongoing to evaluate LT for locally advanced liver-confined non-resectable iCCA that responds to neoadjuvant therapy. Similar to the Methodist study, patients receive gemcitabine-based chemotherapy for 6 months prior to LT. One chemotherapy non-responder also received Y90 SIRT. Preliminary results were recently published, demonstrating an overall transplant rate of 83% (5/6 patients) with 100% OS and 60% RFS with a median follow-up of 15 months [74]. Together, these studies suggest that LT may be considered in select patients with locally advanced disease, and the response to therapy, rather than tumor volume, could identify patients for transplant. 

Given improved responses to therapy offered by alternative chemotherapy regimens, including immunotherapy and targeted therapy, there is increased interest in incorporating these regimens as neoadjuvant therapy in LT. While there is limited published data regarding these treatments prior to LT, two ongoing clinical trials (NCT06140134 [Rutgers, NJ, USA] and NCT06862934 [Milan, Italy]) include patients receiving immunotherapy and/or targeted therapies in neoadjuvant regimens. The viability of such approaches is to be determined. Only a single LT case has been reported for a patient receiving pre-LT immunotherapy. Following tumor response to gemcitabine/cisplatin/durvalumab, the patient received living donor LT with an OS and RFS of >6 months [75]. Targeted therapy with pemigatinib has also been reported in the neoadjuvant LT setting, in a patient who progressed on gemcitabine/cisplatin. After 6 months of metabolic and radiographic response to pemigatinib, the patient received LDLT with no recurrence after 1 year [76].

Incorporation of local therapies into pre-LT transplant regimens is also under investigation. In a retrospective analysis of 30 iCCA patients undergoing LT at UCLA over 30 years, the authors reported that four patients receiving combined systemic and liver-directed therapy had a 100% OS and RFS at 5 years [77]. Another intent-to-treat analysis evaluating Gem/Cis followed by TARE reported that five-year OS was 100% for the 4/17 who completed the protocol and received transplant, compared with 0% for the 13 patients who did not receive transplant [78]. Finally, a recent case report described a patient with multifocal advanced iCCA who received TARE, gemcitabine/cisplatin, and FOLFOX (folinic acid, fluorouracil, and oxaliplatin) prior to LT. Despite having a large multifocal tumor with lymphovascular involvement, the patient had a sustained treatment response prior to LT and remained recurrence-free 16 months post-transplant [79]. While these case series are limited, combined liver-directed and systemic therapy may optimize LT outcomes for locally advanced iCCA, and improved systemic options may further impact survival. Outcomes for studies evaluating LT for iCCA are summarized in Table 3. Ongoing clinical trials evaluating LT for iCCA are listed in Table 4. 

Based on expanding evidence supporting the benefit of LT for iCCA, policy recommendations have recently been updated. In 2023, the AASLD updated its recommendation from “not recommended” to “consideration under research protocols” [87]. The European Association for the Study of the Liver (EASL) and the International Liver Cancer Association (ILCA) similarly recommend LT considerations for early-stage iCCA ≤3 cm [88]. Until recently, the United Network of Organ Sharing (UNOS) granted MELD exception (median MELD at transplant -3, MMaT) for pCCA but not iCCA; however, in June 2024, the OPTN introduced an updated guidance document allowing MELD exception for select iCCA. The current policy, which became active in February 2025, grants exception to patients with underlying cirrhosis who have a solitary unresectable iCCA lesion ≤3 cm and who have demonstrated tumor response or stability for >6 months [89]. While this policy will improve transplant access to patients with iCCA, the strict selection criteria will likely restrict benefit; future policy modifications allowing transplants with greater disease burden may be warranted as evidence increases. A timeline summarizing how indications have evolved for LT in iCCA is summarized in Figure 1.

Careful patient selection is critical in optimizing outcomes following LT for iCCA patients. Disease stability and response to neoadjuvant treatment have been used as surrogate indicators of favorable tumor biology [71,78]. Favorable survival for patients with significant disease burden in case series from Houston Methodist and MD Anderson suggests that tumor biology, rather than size, might indicate patients who would benefit from transplant [71]. These complex factors include genetic and epigenetic alterations (e.g., chromatin instability, DNA repair defects, loss of tumor suppressor genes, oncogene activation, aneuploidy, histone modifications, and DNA methylation), the tumor microenvironment, infiltration and activation of immune cells, cellular differentiation, tumor metabolic alterations, and genomic heterogenicity, hormonal and growth factor signaling, and immune evasion techniques by the tumor. Biologic factors are difficult to ascertain radiographically or on biopsy, leaving clinical response to therapy as the current mainstay of assessment of biologic favorability. In the future, genomic profiling, biomarker identification, circulating tumor DNA, and multiplex transcriptional and proteomic assessments may aid clinical decisions, but current data do not yet support implementation.

The role of liver transplantation for iCCA patients is a subject of ongoing investigation, but emerging data indicate that it can provide a sustained survival benefit to select patients. Our center’s opinion is that current data support expanded implementation of LT for iCCA for unresectable patients with no extrahepatic disease and disease stability for at least 6 months. The decision to offer LT needs to balance ethical issues such as organ utilization, organ availability, and predicted outcomes. In the United States, organ demand outpaces available livers for transplant, which limits the availability of deceased donors to candidates outside of the MELD exception guideline. In some European countries, deceased donor organ availability exceeds demand. We may see more aggressive expansion of iCCA transplant protocols in these locations. Given the impressive outcomes recently noted for more locally advanced iCCA, it is likely that the current UNOS selection criteria are overly restrictive. Due to the rarity of iCCA, multi-institutional or multinational prospective clinical trials would be necessary to definitively determine optimal criteria for LT. Such trials are difficult due to funding limitations, differences in center practices, and logistics. As such, an international registry to collect granular data regarding patient outcomes, radiographic and pathologic characteristics, and treatments will be necessary to obtain to more definitely determine criteria.

One alternative to expanded organ availability in iCCA is the use of living donor liver transplantation (LDLT). LDLT has been widely adopted for HCC, and is not restricted to UNOS exception criteria. As a result, several centers preferentially use LDLT for oncologic indications, such as pCCA, iCCA, and colorectal liver metastases. LDLT also allows better control of transplant timing relative to therapy, which benefits patients receiving immunotherapy or radiation, may reduce waitlist drop-out due to limited organ availability, and reduces graft ischemia-reperfusion injury. Given the risk to the healthy living donor, some feel that use of a living donor for an indication outside of current guidance is not warranted [90].

Currently, patient candidacy for transplant for iCCA is based on center-specific criteria regarding size, prior treatment, and disease stability. The rare nature of the disease and strict selection criteria may limit qualifying candidates and delay data collection regarding optimal benefit. Often, patients are referred to specialized centers only after failing multiple rounds of therapy, limiting candidacy for aggressive approaches, such as downstaging for resection or liver transplant. Early referral should be prioritized and, given the complex and evolving nature of this disease, specialized multidisciplinary patient assessment and management should be preferred.

## 7. Conclusions

The treatment landscape for unresectable locally advanced intrahepatic cholangiocarcinoma continues to evolve, driven by research in systemic and targeted therapies, locoregional treatments, and liver transplantation. Gemcitabine and cisplatin chemotherapy remain the standard first-line therapy for unresectable disease, but emerging immunotherapies and targeted therapies offer the potential for improved survival outcomes. Locoregional treatments, including radiation, TACE, TARE, and hepatic artery infusion, provide options for unresectable patients in select cases, and these treatments may expand access to definitive curative treatments such as surgical resection and liver transplant. A multidisciplinary approach and identifying treatment options with next-generation sequencing is vital for optimizing outcomes. Future research into targeted therapies and transplantation offers hope for improving survival and quality of life in patients with this aggressive malignancy.

## Figures and Tables

**Figure 1 curroncol-32-00293-f001:**
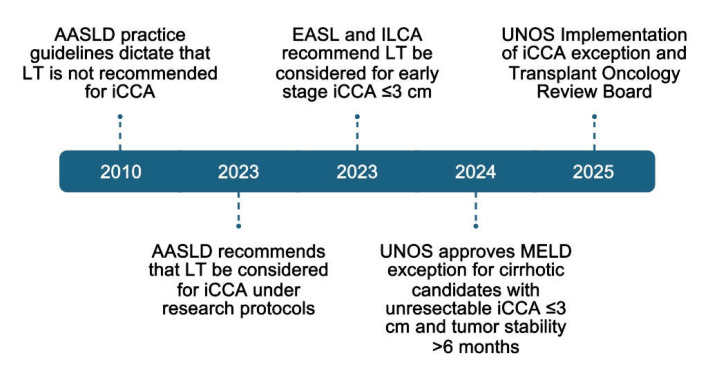
Timeline of evolving indications for LT for iCCA [87,88,89].

**Table 1 curroncol-32-00293-t001:** Prospective trials investigating neoadjuvant regimens for iCCA.

Reference	Intervention	Study	Status	Endpoint	Results
Maithel 2023 [28]	Gemcitabine/cisplatin/nab-paclitaxel + resection	Multi-institutional, phase II	Complete	Primary: completion of both preoperative chemotherapy + resection Secondary: AEs, radiologic response, RFS and OS	Primary: 30 completed preoperative chemotherapy and 22 were resected Secondary: 90% with disease control, 23% with partial response Median OS 24 months Median RFS 7.1 months
NCT03579771 [29]	Gemcitabine/cisplatin/nab-paclitaxel + FGFR2 inhibitor (for patients with FGFR2 fusion or rearrangement) + resection	Single-arm, phase II	Active, notrecruiting	Primary: completion of all preoperative + operative therapy, safety Secondary: radiological response, RFS, OS	-
NCT06050252 [30]	Gemcitabine/cisplatin/durvalumab + resection	Multi- institutional, phase II	Actively recruiting	Primary: completion rate of neoadjuvant treatment + resection Secondary: major pathologic response	-
NCT05967182 [31]	Gemcitabine/ cisplatin/ pembrolizumab + resection	Single-arm, phase II trial	Actively recruiting	Primary: RFS and major pathologic response	-
NCT06569225 [32]	Gemcitabine/ cisplatin/ nab-paclitaxel/ rilvegostomig + resection	Multi- institutional, phase II	Not yet recruiting	Primary: major pathologic response Secondary: radiologic objective response rate, AEs, rate of R0	-

**Table 2 curroncol-32-00293-t002:** Comparison of OS and PFS in prospective trials for liver-directed therapies for unresectable iCCA.

Reference	Intervention	Study	*n*	Median OS	Median PFS
Hong 2016 [37]	High-dose hypofractionated proton beam therapy	Phase II, multi-institutional single-arm study	37	22.5 months (95% CI 12.4–49.7)	8.4 months (95% CI 5.0–15.7)
Martin 2022 [45]	DEBIRI TACE + Gem/Cis	Phase II, multicenter randomized study	48	33.7 months (95% CI 13.5–54.5)	31.9 months (95% CI 8.5–75.3)
Edeline 2020 [46]	TARE + Gem/Cis	Phase II, multicenter clinical trial	41	22 months (95% CI 14–52)	14 months (95% CI 8–17 months)
Chan 2022 [48]	TARE + Gem/Cis	Phase II, multicenter single-arm clinical trial	16	21.6 months (95% CI 7.3–25.2)	9 months (95% CI 3.2–13.1)
Cercek 2020 [53]	HAI floxuridine + systemic Gem/Ox	Phase II, single-arm clinical trial	38	25 months (95% CI, 20.60 not reached)	11.7 months (1-sided 95% CI 11.1)

**Table 3 curroncol-32-00293-t003:** Comparison of outcomes for studies evaluating LT for iCCA.

Reference	Study Design	Population	Intervention	Key Outcomes
Penn 1991 [61]	Retrospective	iCCA patients	LT	2- and 5-yr OS: 30%, 17% Recurrence: 44%
Sapisochin 2014 [63]	Retrospective multicenter	29 iCCA patients with cirrhosis (8 with “very early” iCCA ≤ 2 cm)	LT	In “very early” subgroup: Recurrence: 0% OS 1-, 3-, 5-yr: 100%, 73%, 73%
Sapisochin 2016 [64]	Retrospective multicenter	“Very early” iCCA (≤2 cm) vs. advanced iCCA (>2 cm or multifocal)	LT	Recurrence at 1, 3, and 5 yrs: 7%, 18%, 18% (“very early”) vs. 30%, 47%, 61%; OS: 93%, 84%, 65% vs. 79%, 50%, 45%
De Martin 2020 [65]	Retrospective multicenter	Cirrhotic patients with iCCA or combined HCC and cholangiocarcinoma ≤ 5 cm	LT vs. resection	5-yr RFS: 75% (LT) vs. 36% (resection) For tumors 2–5 cm: recurrence 21% (LT) vs. 48% (resection); RFS 74% vs. 40%
Hong 2011 [70]	Retrospective	iCCA and pCCA patients	LT ± neoadjuvant/adjuvant therapy	5-yr RFS: 47% (neoadjuvant + adjuvant) vs. 33% (adjuvant only) vs. 20% (none)
Lunsford 2018 [71]	Prospective case series	Unresectable iCCA, 6 patients	LT after >6 months disease stability	1-, 3-, 5-yr OS: 100%, 83.3%, 83.3% RFS: 50%
McMillan 2022 [72]	Prospective case series follow-up	Unresectable iCCA, 18 patients	LT after >6 months disease stability	1-, 3-, 5-yr OS: 100%, 71%, 57% 1-, 3-yr RFS: 72%, 52%
Semaan 2025 [73]	Retrospective single-center	Unresectable iCCA, 26 patients	Neoadjuvant treatment + LT	1-, 3-yr OS: 96%, 82.7% 1-, 3-yr RFS: 70.8%, 56.3%
Yaqub 2025 [74]	Prospective single-center	Unresectable locally advanced iCCA with prior response to neoadjuvant therapy	LT	5 patients underwent LT 2 had recurrence at 12 and 13 months
Teixeira 2024 [79]	Case report	Unresectable locally advanced iCCA	Y90 TARE + Gem/Cis + FOLFOX + LT	Recurrence-free at 16-month follow-up
Fernandes 2024 [75]	Case reports	Unresectable locally advanced iCCA	Gemcitabine/cisplatin OR gemcitabine/cisplatin/durvalumab + LT	Recurrence-free at 23-month and 6-month follow-up
Byrne 2025 [76]	Case report	Unresectable iCCA	Y90 TARE + Gem/Cis + Pemigatinib + LT	Recurrence-free at 1-year follow-up
Ito 2022 [77]	Retrospective	30 iCCA patients	LT ± neoadjuvant/systemic + LRT	1-, 3-, and 5-yr OS: 73%, 46%, 42% 1-, 3-, 5-yr OS (for patients transplanted 2008–2019): 100%, 86%, 69%100% 5-yr OS and RFS (for patients treated with systemic + LRT)
Maspero 2024 [78]	Prospective single-center	13 iCCA patients, 4 patients transplanted	Gem/Cis + TARE + LT	5-yr OS: 100% (LT) vs. 0% (no LT)

**Table 4 curroncol-32-00293-t004:** Clinical trials in liver transplant for iCCA.

Reference	Location	Description	Neoadjuvant	Study Type	Status
NCT02878473 [80]	Toronto, Canada	5-year overall survival; LT for pts with cirrhosis and unresectable iCCA ≤2 cm confirmed by biopsy	None or LRT	Multicenter clinical trial, not randomized, phase 2	Terminated
NCT04195503 [81]	Toronto, Canada	5-year overall survival; LDLT for locally advanced unresectable iCCA with no distant mets, LN, or vascular invasion	>6 months stability CTX Alone	Prospective single-center	Recruiting
NCT04556214 [82]	Oslo, Norway	5-year overall survival; LT for locally advanced unresectable iCCA with no distant mets, LN, or vascular invasion	>6 months stability CTX or LRT	Prospective single-center	Recruiting
NCT06140134 [83]	New Jersey, USA	5-year overall survival; LT for locally advanced unresectable iCCA with no distant mets, LN, or vascular invasion	>6 months stability CTX ± IO ± TARE	Multicenter clinical trial, not randomized, phase 2	Recruiting
NCT06098547 [84]	Padova, Italy	3-year overall survival; with matched retrospective comparison to CTX alone; LT for locally advanced unresectable iCCA with no distant mets, LN, or vascular invasion	>6 months stability CTX Alone	Prospective single-center	Recruiting
NCT06862934 [85]	Milan, Italy	3-year overall survival; LT for locally advanced unresectable iCCA with no distant mets, LN, or vascular invasion	>6 months stability CTX+IO+TARE	Prospective single-center	Recruiting
NCT06539377 [86]	Jena, Germany	5-year overall survival; LDLT for locally advanced unresectable G1/G2 iCCA or HCC/CCA unresectable iCCA with no distant mets, LN, or vascular invasion	>6 months stability CTX + TARE	Prospective single-center	Not yet recruiting

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
