# Peer review of "Therapeutic Advances in Initially Unresectable Locally Advanced Intrahepatic Cholangiocarcinoma: Emerging Treatments and the Role of Liver Transplantation"

_curroncol, 2025, doi:10.3390/curroncol32060293_

Round 1
Reviewer 1 Report
Comments and Suggestions for Authors
Overall this an interesting and timely manuscript.
-The authors should talk a bit more about the importance of biology when selecting iCCA candidates for LT.
-The authors should mention that exception points for small iCCA have now been officially approved. Here is a reference.
Guidance to Liver Transplant Programs and the National Liver Review Board for: Adult MELD Exceptions for Transplant Oncology.
https://optn.transplant.hrsa.gov/media/xpdfswdv/nlrb-guidance_adult-transplant-oncology_feb-2025.pdf
Accessed: 4/22/2025
-The authors should expand a bit on what they feel “the role” of LT for iCCA is. Should this be for patients only with prolonged non-progression time? Should these cases all be in a registry? Should we use living donor or deceased donor grafts?
-Add a table with the transplant studies.
-The authors should briefly discuss LT for mixed HCC-iCCA. Here is a reference.
Dageforde LA, Vachharajani N, Tabrizian P, Agopian V, Halazun K, Maynard E, Croome K, Nagorney D, Hong JC, Lee D, Ferrone C, Baker E, Jarnagin W, Hemming A, Schnickel G, Kimura S, Busuttil R, Lindemann J, Florman S, Holzner ML, Srouji R, Najjar M, Yohanathan L, Cheng J, Amin H, Rickert CA, Yang JD, Kim J, Pasko J, Chapman WC, Majella Doyle MB. Multi-Center Analysis of Liver Transplantation for Combined Hepatocellular Carcinoma-Cholangiocarcinoma Liver Tumors. J Am Coll Surg. 2021 Apr;232(4):361-371.
Reviewer 2 Report
Comments and Suggestions for Authors
This is a nice review by the authors on recent developments in management of Intrahepatic cholangiocarcinoma.
Major points: 1. The outcomes of patients who received newer noeadjuvant chemotherapy before resection can be expanded and possibly a table included.
2. Based on the title, additional information about the recent publications including from the authors' group, should be included regarding the different protocols used by different centers as well as the trials in progress.
3. With the NLRB allowing MMAT -3 for less than 3 cm with stability for 6 mths, options for larger tumors with response and stability (Extended criteria grafts) or living donor grafts can be added to the discussion.
Round 2
Reviewer 1 Report
Comments and Suggestions for Authors
The authors have addressed the reviewer and editor comments and have strengthened the manuscript. No further comments.